# A Universal Aerosol Composition Analysis Method for Optical Tweezers Measurement and Its Application to Determine Hygroscopic Growth Factor of Single-particle Aerosol

Chengyi Fan[1], Chunsheng Zhao[1]

[1]Department of Atmospheric and Oceanic Sciences, School of Physics, Peking University, Beijing 100871, China

*Correspondence to*: Chunsheng Zhao (zcs@pku.edu.cn)

**Abstract.** Traditional hygroscopicity bulk measurements of aerosols using humidified tandem differential mobility analyzer (HTDMA) are limited to population-averaged properties, potentially overlooking individual particle growth processes. Although aerosol optical tweezers enable single-particle measurements, no universal and accurate method exists to determine particle dry radius and hygroscopic growth factor (GF). Here, we develop a robust method using optical tweezers to quantify GF of individual particles accurately. Solution densities were accurately predicted via apparent molar volume, and refractive indices were predicted using the molar refraction method. By fitting particle radius and refractive index across multiple relative humidities under the conservation of solute mass, we retrieve dry particle size and hygroscopic growth curves. Application to typical aerosols, including ammonium sulfate, sodium chloride, and sucrose, yields GF in excellent agreement with reported values and thermodynamic models, while extension to mixed-component particles also demonstrates broad applicability. This study provides the first accurate characterization of single-particle hygroscopic growth with optical tweezers, yielding a self-consistent set of physical parameters and a framework to test and refine thermodynamic models, while improving the representation of aerosol–radiation–cloud interactions in climate models.

## 1 Introduction

Aerosol–water interaction is fundamental to the Earth's climate system, influencing cloud microphysics, atmospheric radiation, climate feedbacks, and multiphase chemistry (Boucher et al., 2013; Kreidenweis and Asa-Awuku, 2014). The hygroscopicity of aerosols—the ability to take up water from ambient air—directly alters their size, optical properties, reactivity, and atmospheric lifetime (Tang et al., 2019). Consequently, hygroscopic growth affects climate both through direct scattering and absorption of solar radiation and through activation into cloud condensation nuclei (CCN), thereby influencing the Earth's radiation budget and modulating cloud formation (Wall et al., 2022; Pariyothon et al., 2023; Pöhlker et al., 2023). Moreover, aerosol hygroscopicity can influence particle aging, the formation of secondary pollutants, and deposition in the human respiratory tract, thereby impacting environmental quality and human health (Lee et al., 2012; Vu et al., 2015).

Despite its importance, accurately characterizing aerosol hygroscopicity remains a major challenge due to the extreme complexity of ambient particles, which span wide size distributions, diverse chemical compositions (inorganic salts, organics,

black carbon, etc.), and heterogeneous mixing states (Yao et al., 2022; Li et al., 2025). Therefore, to fully understand aerosol hygroscopicity, it is essential to track the dynamic evolution of particle size with RH under well-controlled conditions. During the measurement and characterization process, Hygroscopic growth is commonly quantified by the diameter growth factor (GF), mass growth factor ($GF_{mass}$), and the hygroscopicity parameter ($\kappa$), which link dry and humidified particle properties (Petters and Kreidenweis, 2007; Tang et al., 2019).

Several experimental techniques have been developed to probe aerosol hygroscopicity. Humidified tandem differential mobility analyzer (HTDMA) is widely used to provide population-averaged growth factors of accumulation mode aerosol in field campaigns, which is useful for climate model evaluations (Tang et al., 2019). However, HTDMA provides only bulk averages, potentially obscuring key hygroscopic details in chemically complex aerosols, and is further limited by assumptions of sphericity, short residence times, and semi-volatile losses, introduced additional measurement uncertainties (Shingler et al.,

2016). Electrodynamic balances (EDB) is a single particle measurement technique that measures particle mass from equilibrium voltage and retrieves size and refractive index from scattering patterns (Tang and Munkelwitz, 1994). Despite its accuracy, the method is restricted to large (~20 µm), charged particles and suffers from density assumptions and iterative fitting errors.

    To address these limitations, aerosol optical tweezers have emerged as a complementary tool for single-particle

hygroscopicity studies (Qiu et al., 2024). A tightly focused laser beam can stably trap a single particle without requiring net charge, while simultaneously enabling Raman spectrum for chemical composition analysis (Ashkin et al., 1986). Moreover, whispering gallery modes (WGMs) superimposed on the background Raman spectrum and Mie resonance fitting provide highly accurate retrievals of particle radius and refractive index under different RHs (Preston and Reid, 2013). Nevertheless, a key limitation of optical tweezers is the absence of a reliable dry-size reference, since trapped particles must be liquid and

spherical, preventing direct determination of their initial dry radius. Previous attempts have relied on empirical approximations, such as assuming half the wet radius at 80% RH for sea salt in Qiu et al. (2024), but such approaches lack universal applicability and accuracy (Hargreaves et al., 2010). Given optical tweezers' accurate size measurements (~10 nm), high temporal resolution (~1 s), controllable environment, and chemical insights from Raman spectroscopy, developing a universal method to accurately determine dry particle size and hygroscopic growth factors is highly warranted.

In this work, we develop a refractive-index-constrained (RIC) retrieval method that enables robust determination of dry particle size and hygroscopic growth curves from optical tweezers measurements. Using sodium chloride, ammonium sulfate, and sucrose as representative inorganic and organic aerosols, we combine density and refractive index data with molar refraction theory to constrain solute mass in particles. This approach yields internally consistent estimates of dry radius, refractive index, density, and mass across RH conditions. The measured hygroscopic growth factors agree well with literature

values, demonstrating the validity of our method. Our study establishes the first systematic and universal framework for

quantifying aerosol hygroscopicity with optical tweezers, offering a pathway to test thermodynamic models (e.g., Köhler theory, Zdanovskii–Stokes–Robinson mixing rules) and to advance single-particle analyses of aerosol chemistry and microphysics. Beyond the methodological advance, this framework can also improve predictions of aerosol climate effects in climate models, owing to the accurate characterization of aerosol hygroscopicity and its strong extensibility.

## 2 Methodology

### 2.1 Optical Tweezers System and Sample Materials

The aerosol optical tweezers system used in this study has been described previously (Fan et al., 2025; Qiu et al., 2024), here we provide a brief summary. As shown in Fig. 1, a 200 mW Gaussian beam from a semiconductor laser (Laser Quantum, Opus-6000, 532 nm) was collimated, expanded, and focused through a high-numerical-aperture objective (Olympus UIS2 PlanC N, 100×, 1.25 N.A.). The tightly focused beam formed a stable optical potential well capable of trapping individual aerosol particles with diameters of 6–12 µm inside the sample cell. Smaller particles become unstable in the trap as the optical gradient force is too weak, causing them to escape from the trap. While stably levitated particles, the same trapping laser also served as the excitation source for Raman scattering. The scattered light was collected, passed through optical filters to remove the excitation beam, and directed into a spectrometer (Zolix Omni-300i, 1200 grooves mm$^{-1}$ grating) for detection, enabling subsequent determination of particle size and refractive index.

Aerosol particles were generated using an ultrasonic nebulizer (Yuyue 402AI model) and introduced into the sample chamber. The nebulizer solution was prepared with high-purity chemicals dissolved in ultrapure water (18 MΩ·cm; SIMGEN, Hangzhou SIMGEN Biotechnology Co., Ltd.). Because trapped droplets can achieve hygroscopic equilibrium within the chamber eventually, their equilibrium state is independent of the initial solute concentration. Therefore, only the solute mass ratios were recorded, not the absolute concentrations. Ammonium sulfate ($(NH_4)_2SO_4$, 99.0% AR; Sinopharm Chemical Reagent Co., Ltd.) and sodium chloride (NaCl, 99.5% AR; Shanghai Titan Scientific Co., Ltd.) were used as representative inorganic aerosol components, and sucrose ($C_{12}H_{22}O_{11}$, 99.9% AR; Sinopharm Chemical Reagent Co., Ltd) was selected as the organic component.

The relative humidity (RH) inside the chamber was controlled in real time by adjusting the mixing ratio of dry and humid nitrogen flows via two mass flow controllers (MFCs, Dmass, DFC10-1/4-N2-3000SCCM-B01). Humidity-temperature probes (Rotronic, HC2A-S) were placed at both the inlet and outlet to monitor RH and temperature within the chamber, and an additional probe (Shenzhen Yowexa Sensor System CO., Ltd., DWL-21E) was placed inside the chamber for offline RH calibration. All experiments were conducted at 20 °C and ambient pressure. Considering probe uncertainties and environmental fluctuations, the overall RH uncertainty was estimated to be ±1%.

Trapped spherical droplets inside the chamber acted as high-quality optical microcavities. Under specific wavelength conditions, spontaneous Raman scattered light underwent internal reflection at the droplet interface, forming standing waves. This resulted in sharp, high-intensity peaks superimposed on the broad Raman background, a phenomenon known as WGMs (Fig. S1) (Benner et al., 1980). The WGMs positions are highly sensitive to both particle radius and refractive index. Using the Mie fitting algorithm developed by Preston and Reid (2013), both parameters were retrieved simultaneously by minimizing

the squared error between calculated and measured WGM wavelengths to below $1 \times 10^{-4}$ nm². Accounting for RH perturbations and fitting uncertainties, the typical precision achieved was ~10 nm in particle radius and 0.002 in refractive index. All refractive index values reported in this study were adjusted to 589 nm using the dispersion relation provided by the fitting algorithm.

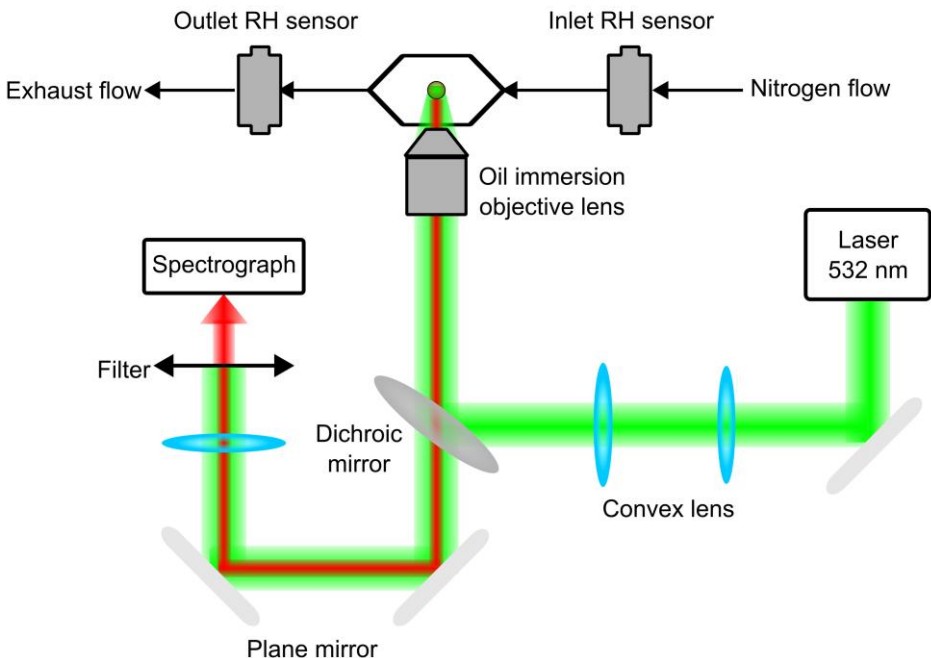

**Figure 1.** Schematic of the aerosol optical tweezers.

## 2.2 Refractive Index and Density Calculations

Before introducing the method for determining the dry particle radius, it is necessary to outline the calculation of refractive index for mixed particles. In this study, we employ the physically based molar refraction method to predict the refractive index of mixtures, rather than relying on the empirical volume-weighted approach (Liu and Daum, 2008; Cai et al., 2016). According

to the molar refraction method,

$$R_e = \left(\frac{n_e^2-1}{n_e^2+2}\right)\frac{M_e}{\rho_e} = \frac{N_A \alpha_e}{3} \tag{1}$$

where $R_e$, $M_e$, and $\alpha_e$ represent the effective molar refraction, molecular weight, and polarizability of the mixture, respectively, all of which are additive on a molar basis. Here, $n_e$ is the refractive index, $N_A$ is Avogadro's number, and $\rho_e$ is the density of the mixture. Cai et al. (2016) demonstrated that when experimental density values are used, the molar refraction method can predict refractive indices with an accuracy of ~0.2%. In contrast, adopting the ideal mixing density assumption—where both mass and volume are treated as additive—introduces errors of up to ~3.5% in density and ~1% in refractive index, with the refractive index error scaling directly with density error. The failure of the ideal mixing density assumption arises from solute–solvent interactions during dissolution (e.g., electrostriction) and the structural modification of water, both of which alter the overall solution volume and thereby cause density deviations (Clegg and Wexler, 2011). Thus, accurate prediction of the refractive index via the molar refraction method requires reliable estimation of the mixture's density.

In this work, solution density is predicted using the apparent molar volume $V_\phi$, which represents the volume increment caused by dissolving one mole of solute in a large amount of solvent (Clegg and Wexler, 2011). At constant temperature, $V_\phi$ is a function of solute concentration (typically molality, $c$, or ionic strength, $I = 1/2 \sum_i c_i z_i^2$, where $c_i$ is the molar concentration of ion $i$ and $z_i$ is its charge). Ionic strength is chosen because it integrates both ion concentration and charge, providing a more accurate measure of the overall electrostatic interaction strength governing non-ideal behavior in electrolyte solutions.

Experimental density data for single-solute solutions, combined with the database of Extended Aerosol Inorganics Model (E-AIM) (Wexler and Clegg, 2002; Clegg and Wexler, 2011; U.S. Department of Agriculture [USDA], 2025), were used to compute $V_\phi$ for each solute via Eq. (2):

$$V_\phi = \frac{M_s}{\rho_e} - M_w \frac{1-x_s}{x_s} \frac{\rho_e-\rho_w}{\rho_e \rho_w} \tag{2}$$

where $M_s$ and $M_w$ are the molecular weights of solute and water, $x_s$ is the solute mole fraction, and $\rho_w$ is the density of water. The derived $V_\phi$ values were further expressed as continuous functions of ionic strength (for inorganics) or molality (for organics) using spline interpolation.

For single-solute solutions, the solution density can be computed by substituting the solute-specific $V_\phi$ into Eq. (3):

$$\rho_e = \frac{m_t}{V_t} = \frac{(1-x_s)M_w + x_s M_s}{(1-x_s)V_w + x_s V_\phi(I \, or \, c)} \tag{3}$$

where $m_t$ and $V_t$ are the total mass and volume of the solution. For multi-solute mixtures, Young's rule is applied to account for the combined contribution of different solutes to the total volume (Young and Smith, 1954; Clegg and Wexler, 2011). The core idea is that the $V_\phi$ of a mixture can be approximated by the weighted average of $V_\phi$ of its components at the same total ionic strength. Meanwhile, for mixtures containing both inorganic salts and organics, the contributions of each are calculated independently. After simplification, the density of a multi-component solution is given by:

$$\rho_e = \frac{(1-\sum_i x_{s,i})M_w + \sum_i x_{s,i} M_i}{(1-\sum_i x_{s,i})V_w + \sum_i x_{s,i} V_{\phi,i}(I \, or \, c)} \tag{4}$$

where the subscript *i* denotes the *i*-th solute.

As shown in Fig. S2, validation against experimental data shows that, for inorganic salt systems, the method predicts the density of ammonium nitrate–ammonium sulfate mixtures with a mean error of only 0.1% (Che et al., 2012). While for inorganic–organic systems such as ethanol–ammonium sulfate, the maximum error is below 1% (Hervello and Sánchez, 2007). Applying the molar refraction method further yields refractive index errors below 0.5% (Urréjola et al., 2010). Overall, this integrated framework achieves prediction accuracies of better than 1% for density and 0.5% for refractive index, providing a robust basis for subsequent hygroscopicity retrievals.

**2.3 Determination of Dry Particle Radius and Hygroscopic Growth Factor**

In this study, refractive index data of single-solute solutions at different solute mass fractions reported in the literature were taken as reference values (Tan and Huang, 2015; Urréjola et al., 2010; USDA, 2025). Assuming the molar refraction of a solute to be $R_0$, we used the functional relation between apparent molar volume and solute concentration together with the molar refraction method to calculate the refractive index at different solute mass fractions. As shown in Fig. S3, by constraining $R_0$ to minimize the total squared errors between calculated and reference values, we determined the molar refractions of individual solutes: 9.33 for NaCl, 23.52 for $(NH_4)_2SO_4$, and 70.22 for sucrose. For water, the molar refraction was calculated using Eq. (1), yielding 3.71.

For a single particle stably trapped in the chamber, the solute mass remains constant during hygroscopic equilibration because the solute is non-volatile. In this work, we varied the RH in the chamber and measured the particle radius and refractive index ($n_{mea}$) at equilibrium under each RH. At each stable RH, we measured the particle for at least 1000 s to obtain the averaged radius and refractive index at that humidity (as shown in Fig. S4). In addition, under our experimental RH conditions, the particle radius changed almost simultaneously with RH, indicating that the particle indeed reached hygroscopic equilibrium at the stabilized RH. For single-solute particles, we assumed the solute mass to be $m_0$ and adopted the molar refraction obtained from solution data. Using Eq. (3) and Eq. (5), the solute mass fraction and density at different particle radii were calculated as:

$$\rho_e(\phi_s, V_\phi(I(\phi_s) \text{ or } c(\phi_s))) \cdot \frac{4}{3}\pi r^3 \cdot \phi_s = m_0 \tag{5}$$

The corresponding refractive index $n_{cal}$ was then computed with Eq. (1). By constraining $m_0$ to minimize the total error ($\sum_j \frac{(n_{cal,j} - n_{mea,j})^2}{\sigma_j^2}$, *j* denotes the *j*-th measurement data point), we obtained the solute mass of the particle, which is referred to as the refractive-index-constrained retrieval method. The dry particle volume and radius were subsequently derived from the crystallized solute density, and hygroscopic growth factors at different RH were determined. It is important to note that the dry particle refers to a particle containing no water, at which point it is fully crystallized and its density and refractive index

correspond to those of the pure solid. Data from both humidification and dehumidification cycles are jointly used to constrain the dry-particle mass. Moreover, at any given RH, the particle radius obtained during humidification and dehumidification agrees within the measurement uncertainty, as shown in Fig. S4(b). Therefore, the direction of RH change does not influence

the retrieved dry particle size or the calculated growth factors. Besides the volume growth factor, we also calculated the mass growth factor to facilitate comparison with literature values.

Since the measured particle radius and refractive index inevitably involve uncertainties, we assumed that the measurements follow a normal distribution characterized by their mean and standard deviation, and performed Monte Carlo sampling from this distribution. Normal distribution is chosen due to the detection noise and stochastic fitting processes. A

175 total of 10,000 samples were generated and constrained to obtain the mean and standard deviation of $m_0$. We also tested simultaneous retrieval of $m_0$ and $R_0$, and the resulting $R_0$ agreed with values derived from solution data within the error margin, consistent with Tang and Munkelwitz's (1994) observation on the stability of molar refraction in supersaturated states and the validity of the molar refraction method.

For multi-solute particles, the solute composition was assumed to match that of the pre-prepared solution. Based on known

mass ratios, the mass of one solute was fixed, and Eq. (4) and Eq. (5) were applied to calculate solute fractions and particle density at different radii. The refractive index was then computed, and RIC method was used to determinate of the dry radius and hygroscopic growth factor of the particle.

## 3 Results and discussion

### 3.1 Measurement of Hygroscopic Growth Factors of Single-Solute Particles

We first validated the accuracy of RIC method using ammonium sulfate as a standard reference particle. As shown in Fig. 2(a), the apparent molar volume as a function of ionic strength provides the basis for calculating the particle density. For a representative ammonium sulfate particle, we measured the radius and refractive index across 65–95% RH and obtained a constrained dry radius of 2.99 µm (± 0.014 µm) using RIC method. Data at lower RH were not measured because particles shrink as RH decreases, eventually becoming too small (approximately 3-4 µm) to be stably captured by the optical tweezers.

Fig. 2(b) compares the measured refractive indices with those calculated using the constrained optimal solute mass, showing a maximum deviation of less than 0.002. Additionally, the error bars for the optical tweezers measurements represent the standard deviation, obtained either directly from the statistics of measurements or calculated through error propagation. Fig. 2(c) presents the GF of ammonium sulfate, together with a fit using Eq. (6):

$$GF = \left(1 + (a + ba_w + ca_w^2)\frac{a_w}{1-a_w}\right)^{1/3} \tag{6}$$

which agrees well with predictions from the thermodynamic E-AIM model within the experimental uncertainty. Comparisons

with HTDMA and EDB measurements further confirm the reliability of the proposed method, as shown in Fig. 2(d) for GF$_{mass}$

of ammonium sulfate (Tang and Munkelwitz, 1994; Zardini et al., 2008). Similar consistency is also observed for sodium

chloride, another major inorganic component of atmospheric aerosols, as shown in Fig. S5.

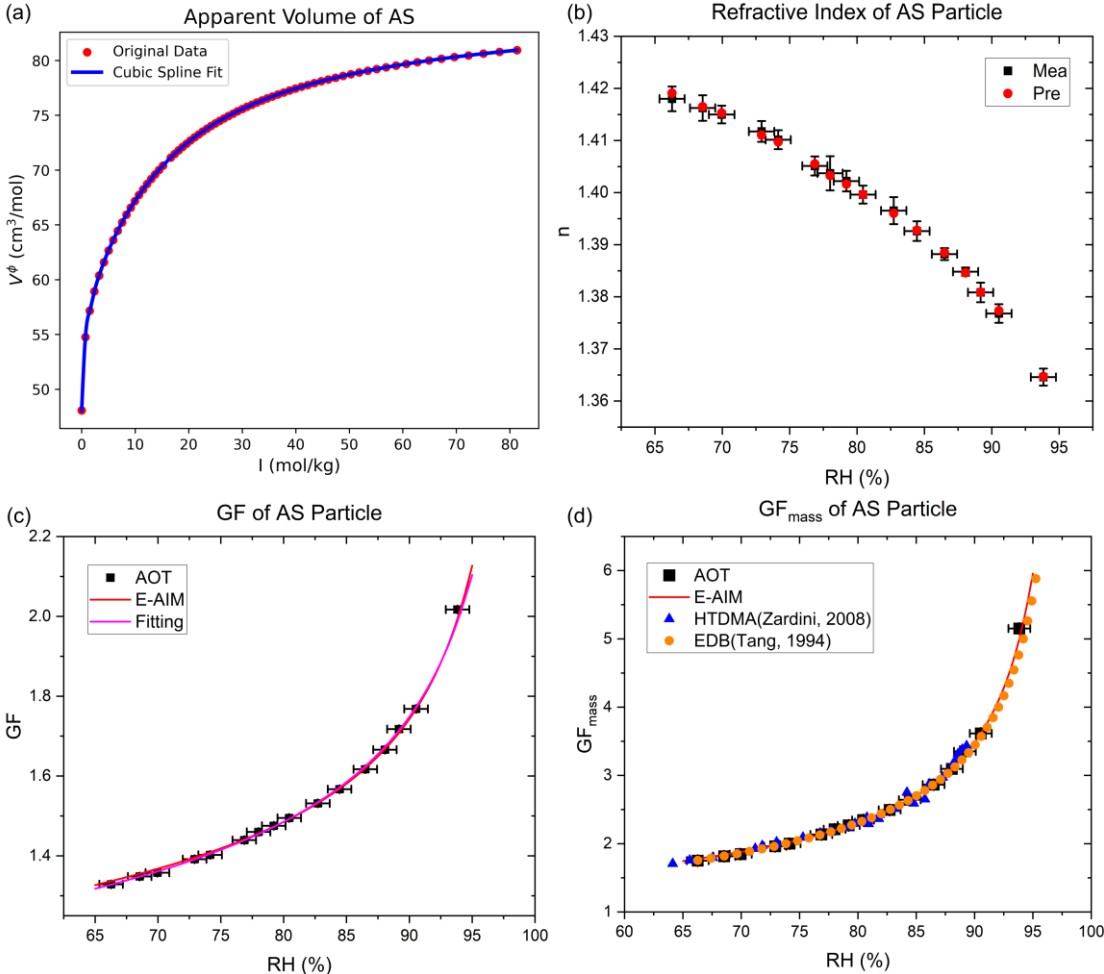

**Figure 2.** Measurement of ammonium sulfate hygroscopicity. (a) Apparent molar volume of ammonium sulfate as a function of ionic strength.

(b) Measured refractive index and corresponding values derived from the constrained solute mass. The error bars for the optical tweezers

measurements represent the standard deviation. (c) Hygroscopic growth factors of ammonium sulfate particles, together with the fitted

growth curve and the E-AIM prediction. AOT means data from aerosol optical tweezers and the standard deviation for growth factors is

sufficiently small that it is largely obscured by the data markers. (d) Mass growth factor of ammonium sulfate compared with E-AIM

predictions, HTDMA measurements (Zardini et al., 2008), and EDB measurements (Tang and Munkelwitz, 1994).

For sucrose, representing organic aerosols, Fig. 3(a) shows the apparent molar volume as a function of molality, derived

from density data of sucrose solutions at different concentrations. We further measured the radius and refractive index of a

typical sucrose particle over 75–95% RH, yielding a constrained dry radius of 3.57 µm ($\pm$0.015 µm). As illustrated in Fig. 3(b),

the measured and calculated refractive indices agree well, with the maximum deviation again below 0.002, demonstrating that

the method is also applicable to organics. Fig. 3(c) compares the GF fitted curve of sucrose particles with literature HTDMA data (Estillore et al., 2017) and the fitting parameters for the GF curves are summarized in Table 1. Good agreement is observed for RH < 85%, whereas at RH > 85%, HTDMA data exhibit systematically lower values and a plateau, while the GF measured by optical tweezers increases rapidly with RH, consistent with the general trend of aerosol water uptake at high humidity. We therefore attribute the HTDMA underestimation to incomplete equilibration or error caused by statistical averaging.

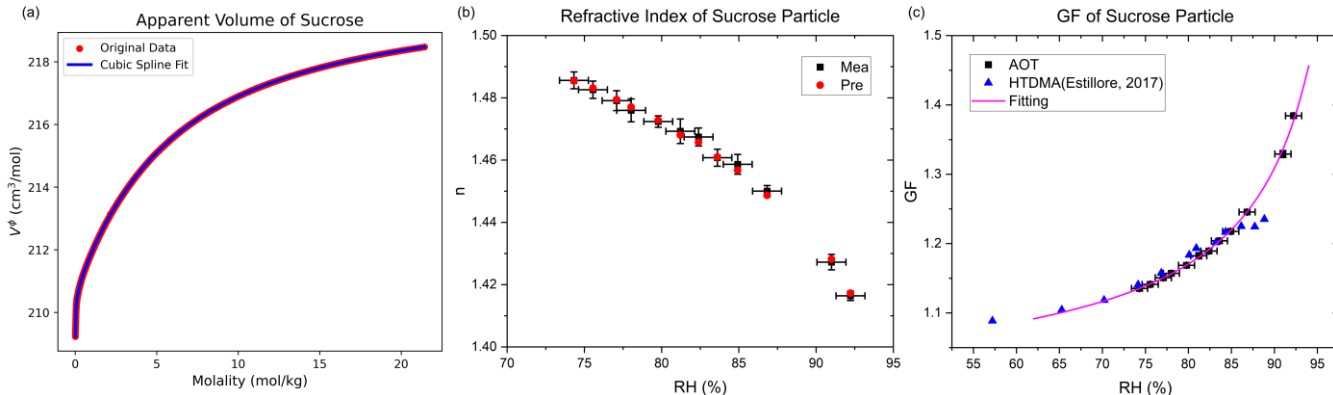

**Figure 3.** Measurement of sucrose hygroscopicity. (a) Apparent molar volume of sucrose as a function of molality. (b) Measured refractive index and corresponding values derived from the constrained solute mass. (c) Hygroscopic growth factor of sucrose with fitted growth curve and HTDMA measurements (Estillore et al., 2017). AOT means data from aerosol optical tweezers.

**Table 1.** Summary of fitting parameters for the GF curves.

| Parameters | Ammonium Sulfate (AS) | Sodium Chloride | Sucrose | AS&NaCl (mass ratio = 1:1) |
|:---:|:---:|:---:|:---:|:---:|
| $a$ | 1.158 | 4.171 | 0.394 | -0.622 |
| $b$ | -0.620 | -2.964 | -0.456 | 5.512 |
| $c$ | -0.146 | -0.045 | 0.190 | -4.266 |

**3.2 Measurement of Hygroscopic Growth Factors of Multi-Solute Particles**

We further applied RIC method to multicomponent aerosols. A mixed solution of ammonium sulfate and sodium chloride at a 1:1 mass ratio was nebulized to generate aerosol particles, which were subsequently trapped by optical tweezers. The mass ratio of ammonium sulfate to sodium chloride in the trapped particles was assumed to remain unchanged. For a representative mixed particle, the radius and refractive index were measured over 65–95% RH, yielding a constrained dry radius of 3.03 µm (±0.012 µm). As shown in Fig. 4(a), the measured and calculated refractive indices agree closely, with a maximum deviation of less than 0.002. Fig. 4(b) further presents the GF of the mixed particle, along with E-AIM predictions. Good agreement is

observed for RH > 75%, while at RH < 75%, E-AIM results are slightly higher. This discrepancy may arise from the stronger ion–ion and solute–solvent interactions in highly supersaturated droplets, where the empirical basis of E-AIM becomes less accurate. Nevertheless, the overall consistency demonstrates the applicability of our method to multicomponent particles.

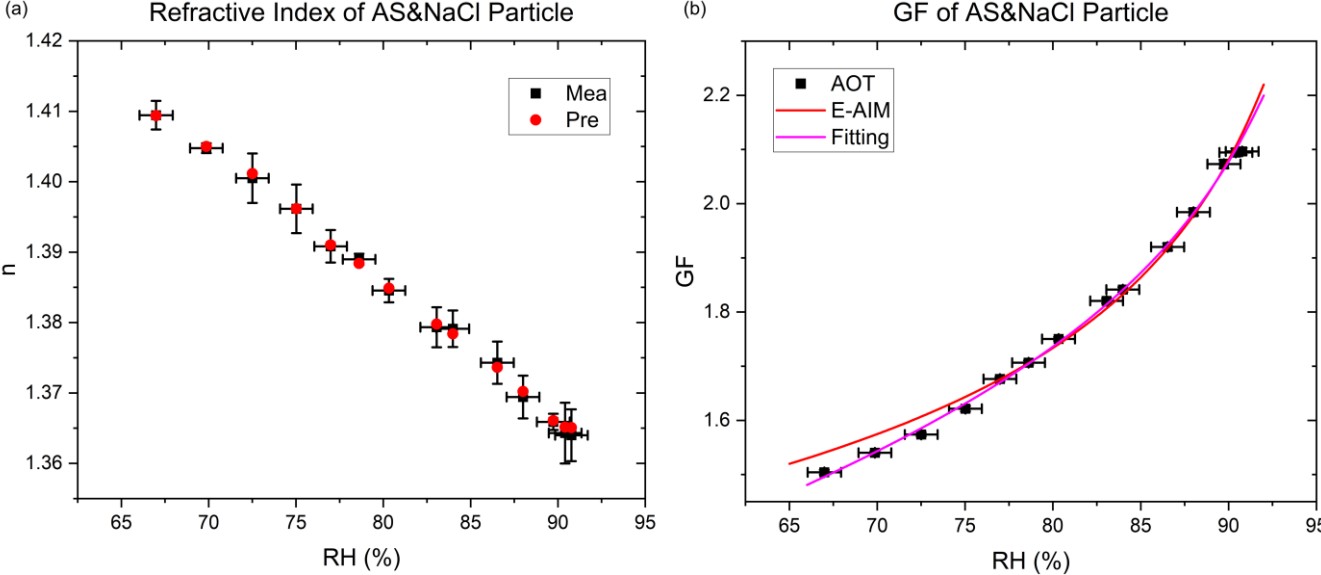

**Figure 4.** Measurement of mixed ammonium sulfate–sodium chloride particles hygroscopicity. (a) Measured refractive index and corresponding values derived from the constrained solute mass. (b) Hygroscopic growth factor of mixed particles compared with E-AIM predictions. AOT means data from aerosol optical tweezers.

It is worth noting that although we assumed the solute mass ratio in mixed particles to be identical to that of the precursor solution, the framework also allows both solute masses to be treated as free variables during constraint retrieval. In practice, the retrieved solute mass ratio remained 1:1 within experimental uncertainty. This suggests that our approach can also be used to test and validate the assumed composition of mixed aerosol particles in future applications.

However, our method is currently applicable only to internally mixed particles. This is because the optical tweezers can trap only liquid droplets, and the retrieval framework requires the particle to be homogeneous. For externally mixed aerosols, insoluble inclusions may be present, leading to a heterogeneous refractive-index distribution. In such cases, both optical trapping stability and the spherical, homogeneous Mie scattering assumption may break down. For these types of particles, techniques such as HTDMA, or the development of Bessel-beam optical tweezers capable of trapping solid particles, would be more suitable for hygroscopicity measurements (Zhao et al., 2020). For particles containing substantial organic material or surfactants, liquid–liquid phase separation (LLPS) may occur at low RH. This would invalidate the standard Mie-fitting procedure, and additional models—such as core–shell Mie calculations—would be required to retrieve the radii and refractive

indices of the individual phases before applying further thermodynamic constraints (Vennes and Preston, 2019). In contrast, if no LLPS occurs, changes in surface tension induced by organics are unlikely to affect the results, because Kelvin effects are negligible for micron-sized droplets.

Although a detailed treatment of these scenarios is beyond the scope of the present study, we suggest that the method could be extended in the future by incorporating more sophisticated optical models (e.g., core–shell Mie theory) as well as trapping techniques compatible with multiphase particles.

### 3.3 Advanced Error Analysis and Applicability of the RIC Approach

In principle, each measurement point at a given RH allows one to calculate a solute mass using the particle's refractive index, radius, and the molar refraction method. However, since both refractive index and radius carry inherent uncertainties, back-calculating the solute mass from a single measurement point and then inferring the GF would introduce large errors. To address this, the present method elegantly exploits the conservation of non-volatile solute mass and applies refractive index constraints across multiple measurement points simultaneously, thereby minimizing errors arising from uncertainties in refractive index and radius.

Idealized calculations show that with typical uncertainties of 0.002 in refractive index and 10 nm in radius, the relative error in retrieving the dry radius of a single-solute particle falls below 0.5% when more than ten measurement points are used, as shown in Fig. 5(a), whereas the error exceeds 2% when relying on a single point. Fig. 5(b) further illustrates how, with radius error fixed at 10 nm, the relative error in dry radius decreases with smaller refractive index error. When the refractive index error is reduced to 0.0005, the relative errors corresponding to the two measurement scenarios fall to 0.2% and 0.7%, respectively, enabling accurate application of this method to particles undergoing interfacial chemical reactions or containing volatile solutes. Fig. 5(c) shows that, with refractive index error fixed at 0.002, the dry radius precision is nearly insensitive to radius error, indicating that refractive index precision plays a dominant role in reducing uncertainty. This is because the proposed RIC method primarily derives particle composition and dry radius from their refractive indices. Consequently, improving the accuracy of refractive index measurements leads to a more pronounced reduction in errors of the inferred dry radius.

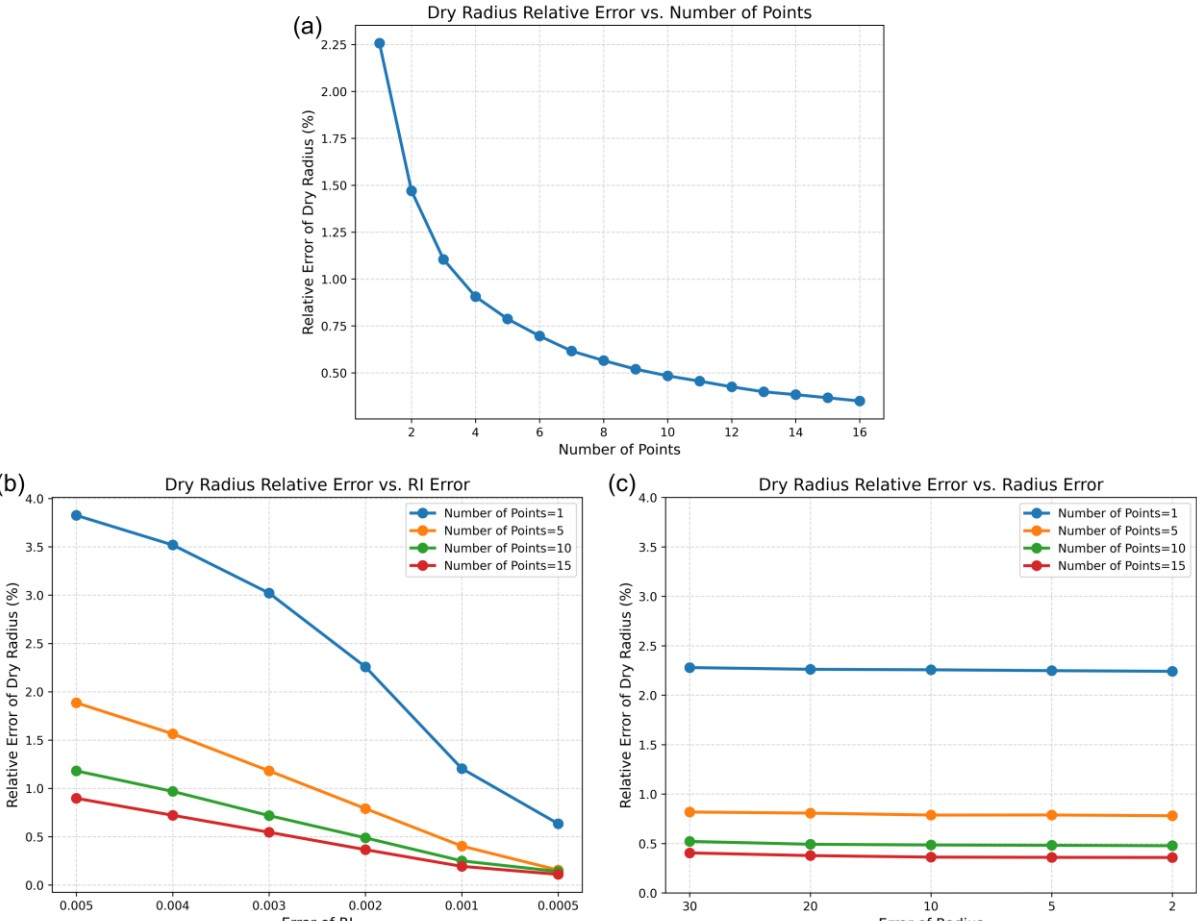

**Figure 5.** Error analysis of the RIC method. (a) Relative error of the dry radius as a function of the number of measurement points, with refractive index and radius errors fixed at 0.002 and 10 nm, respectively. (b) Relative error of the dry radius as a function of refractive index error for different numbers of measurement points, with radius error fixed at 10 nm. (c) Relative error of the dry radius as a function of radius error for different numbers of measurement points, with refractive index error fixed at 0.002.

At present, this framework can constrain solute mass and calculate hygroscopic growth factors for aerosol particles with known composition, showing strong versatility. For particles generated from solutions with unknown components, some limitations remain. Nevertheless, the approach could be extended by adopting Tang and Munkelwitz's (1994) idea of fitting the solution density–concentration relationship with a three-parameter ternary equation (fixing the constant term as pure water density), while treating the unknown solute's molecular weight, molar refraction, and mass as free parameters. In this case, six unknowns can be solved simultaneously through multi-point constraints. Provided sufficient measurement points across RH and very small refractive index errors, this method still holds promise for retrieving both the composition and hygroscopicity of such particles.

**4 Summary and Conclusions**

Aerosol hygroscopic growth critically influences particle size and optical properties, thereby affecting the Earth's radiation balance and climate (Kreidenweis and Asa-Awuku, 2014; Tang et al., 2019). Current measurement techniques, such as HTDMA, primarily provide bulk statistics and may obscure important single-particle processes and details. Although aerosol optical tweezers enable accurate single-particle measurements, a universal method for determining particle dry radius and hygroscopic growth factor is still lacking. In this study, we developed a universal RIC method using optical tweezers to

accurately measure the hygroscopic growth factor of individual aerosol particles. Using apparent molar volume, we first predicted the density of mixed solutions within 1% error, and subsequently applied the molar refraction method to predict refractive indices within 0.5% error. For single trapped particles, the conservation of non-volatile solute mass was combined with measurements of particle radius and refractive index under varying RH to constrain solute mass, enabling accurate determination of dry particle radius and hygroscopic growth factor.

We validated the method using common aerosol components—ammonium sulfate, sodium chloride, and sucrose—and found excellent agreement with literature values and thermodynamic models. Measurements were also successfully extended to multi-component particles, demonstrating the method's broad applicability. Further analysis suggests that, given sufficient measurement points and improved refractive index precision, the approach has potential to resolve the composition and hygroscopic behavior of particles containing volatile solutes or undergoing interfacial chemical reactions. Potential

applications to particles of unknown composition were also discussed.

This work represents the first systematic application of optical tweezers to determine dry particle radius and hygroscopic growth factors with high precision. It provides a self-consistent set of particle physical parameters, including mass, density, and refractive index, and offers a framework for testing and refining thermodynamic models (e.g., Köhler theory, Zdanovskii–Stokes–Robinson mixing rules), improving our understanding and representation of aerosol–cloud and aerosol–radiation

interactions in climate studies.

**Code availability**

Codes used in this study are available on request from the corresponding author (email: zcs@pku.edu.cn).

**Data availability**

Data used in this study are available on request from the corresponding author (email: zcs@pku.edu.cn).

**Author contribution**

CF put forward the idea, performed the experiments, analyzed the data, and wrote the manuscript. C.Z. participated in the discussion, reviewed and revised the manuscript.

**Competing interests**

The authors declare that they have no conflict of interest.

**Acknowledgements**

We gratefully acknowledge the financial support of National Natural Science Foundation of China (42530602) and the National Natural Science Foundation of China (42275070). During the preparation of this work, the authors used ChatGPT to polish the paper. After using this tool, the authors reviewed and edited the content as needed and take full responsibility for the publication's content.

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
