# Peer review of "A Universal Aerosol Composition Analysis Method for Optical Tweezers Measurement and Its Application to Determine Hygroscopic Growth Factor of Single-particle Aerosol"

_EGUsphere, 2025_

## Author Comment (AC1)

Response to Anonymous Referee #1

*The manuscript presents scientifically valuable and timely work on advancing single-particle hygroscopicity measurements using aerosol optical tweezers. The authors propose a novel and robust methodology for retrieving dry particle size and hygroscopic growth factors, and the results show clear potential for improving our understanding of aerosol physicochemical properties and their representation in climate models. While the study is of high scientific relevance and demonstrates promising methodological innovation, the current version of the manuscript requires major improvements in clarity, structure, and methodological justification before it can be considered for publication.*

Response: We would like to express our deepest gratitude for taking the time to review our manuscript. In response to your constructive feedback, we have revised our manuscript. Below, we will provide a point-by-point response to your comments. All the changes have been included in the newest version of our manuscript.

*Please specify the measurement duration at each relative humidity and discuss how it relates to the particle (droplet) relaxation time. This information is essential for assessing whether equilibrium conditions were reached during the measurements.*

Response: Thanks for your valuable advice. At each stable relative humidity (RH), we measured the particle for at least 1000 s to obtain the averaged radius and refractive index at that humidity. Since the particle radius responded almost simultaneously to changes in RH across all experimental RH ranges, we consider the particle to have reached hygroscopic equilibrium within the stable-RH window. This passage has been added to section 2.3:

"At each stable RH, we measured the particle for at least 1000 s to obtain the averaged radius and refractive index at that humidity (as shown in Fig. S4). In addition, under our experimental RH conditions, the particle radius changed almost simultaneously with RH, indicating that the particle indeed reached hygroscopic equilibrium at the stabilized RH."

[Figure]

Figure S4. The radius of an ammonium sulfate particle under varying relative humidity. (a) 93%–91%; the purple shaded box indicates the time period during which RH was stable and the particle radius was averaged. (b) 90%–87%.

*The meaning of the error bars is unclear. Do they represent standard deviation, standard error, or another metric? Please ensure consistency and explain why error bars are included in some cases but omitted in others. In Fig. S2C, the box plot presentation also requires clarification—do the bounds represent specific quantiles? Throughout the manuscript, any statistical tool or metric used should be explicitly defined and described.*

Response: Thank you for raising this point. The error bars in our figures represent

standard deviation (SD). For all measurements obtained directly from the optical tweezers system, we include error bars on both the x-axis and y-axis. Because the uncertainty in RH is relatively large, the SD in RH is clearly visible on the x-axis. For the y-axis, the uncertainties of GF and $GF_{mass}$—computed through error propagation—are small and largely obscured by the marker size, whereas the uncertainty in the refractive index, n is larger and therefore more apparent (e.g., Fig. 2). To avoid confusion, we have clarified this information in both the main text (section 3.1) and the figure captions:

"Additionally, the error bars for the optical tweezers measurements represent the standard deviation, obtained either directly from the statistics of measurements or calculated through error propagation."
"(b) Measured refractive index and corresponding values derived from the constrained solute mass. The error bars for the optical tweezers measurements represent the standard deviation. (c) Hygroscopic growth factors of ammonium sulfate particles, together with the fitted growth curve and the E-AIM prediction. AOT means data from aerosol optical tweezers and the standard deviation for growth factors is sufficiently small that it is largely obscured by the data markers."

For the thermodynamic model results and the literature data shown in the figures, no uncertainty information is available; therefore, no error bars are displayed for those datasets.

In Fig. S2(c), the box plot boundaries represent, from top to bottom: maximum value, third quartile (Q3), median, first quartile (Q1), and minimum value. This has been explicitly described in the figure caption.

"(c) Boxplots of density errors from the two methods. The box plot boundaries represent, from top to bottom: maximum value, third quartile (Q3), median, first quartile (Q1), and minimum value."

***Please explain why the RH range starts at 65%. What prevents measurements at lower humidities?***

Response: Thank you for your valuable suggestion. Data at lower RH were not directly measured because particles shrink as they lose water, eventually becoming too small to be stably captured by the optical tweezers and falling off. Most particle dropped at relative humidities of 60%–70%, corresponding to particle radius of approximately 3–4 µm. This passage has been added to section 3.1:

"Data at lower RH were not measured because particles shrink as RH decreases, eventually becoming too small (approximately 3-4 µm) to be stably captured by the optical tweezers."

***In line 52, the statement "accurate size measurements, high temporal resolution" requires quantification. How accurate are the size measurements, and what is the actual temporal resolution achieved? Please provide numerical values or ranges.***

Response: Thank you for your suggestion. The precision of the optical tweezers measurements for particle radius is approximately 10 nm, with a temporal resolution of about 1 s. This sentence has now been revised to:

"Given optical tweezers' accurate size measurements (~10 nm), high temporal resolution (~1 s), controllable environment, and chemical insights from Raman spectroscopy, developing a universal

method to accurately determine dry particle size and hygroscopic growth factors is highly warranted."

***Have any statistical metrics been applied to evaluate model performance or measurement reliability (e.g., RMSE, confidence intervals, significance tests)? If so, please report them; if not, please justify their absence.***

Response: Thank you for the valuable suggestion. No additional statistical metrics were applied for further evaluation. This is because the thermodynamic model curves and literature data used for comparison are deterministic and contain no stochastic components; therefore, statistical measures such as RMSE or significance testing are not directly applicable. We have already provided the standard deviation for all optical tweezers measurements (though in some cases, such as GF, the SD is too small to be visible). All comparison data fall within the SD range of the optical tweezers results, demonstrating the reliability of our measurements.

***The manuscript discusses dry diameter retrieval extensively, yet the results for truly dry particles are not clearly presented. Since the method is applied mainly to droplets, please clarify how "dry conditions" are defined in practice. Additionally, hysteresis effects should be addressed: does the direction of RH change (humidification vs. dehumidification) influence the retrieved sizes or growth factors? Please elaborate.***

Response: Thank you for your thoughtful comments regarding the definition of dry conditions and the potential influence of hysteresis. In our experiments, the "dry state" refers to the condition in which the particle contains no water, analogous to the pre-deliquescence solid state. Under this condition, the particle is crystallized, and its density and refractive index correspond to those of the pure solid. Using the refractive-index–constrained method, we retrieve the solute mass inside the particle; dividing this mass by the density of the solid solute yields the solute volume, which we treat as the dry particle volume. The dry particle radius is then calculated accordingly.

Because the optical tweezer can trap only liquid droplets, the particles in our measurements undergo humidification and dehumidification without any phase transitions. The refractive-index–constrained retrieval relies on radius and refractive index measured at multiple RH values; moreover, when the particle reaches the same relative humidity during humidification and dehumidification, its radius and refractive index agree within the measurement uncertainty, as shown in Fig. S2(b). Consequently, hysteresis does not affect the inferred dry size or the calculated growth factors. This sentence has been added to section 2.3:

"The dry particle volume and radius were subsequently derived from the crystallized solute density, and hygroscopic growth factors at different RH were determined. It is important to note that the dry particle refers to a particle containing no water, at which point it is fully crystallized and its density and refractive index correspond to those of the pure solid. Data from both humidification and dehumidification cycles are jointly used to constrain the dry-particle mass. Moreover, at any given RH, the particle radius obtained during humidification and dehumidification agrees within the

measurement uncertainty, as shown in Fig. S4(b). Therefore, the direction of RH change does not influence the retrieved dry particle size or the calculated growth factors."

***The generated and analyzed particles (droplets?) are reported to fall within the 6–12 µm size range (line 71). What occurs when particles (droplets?) shrink below this range? Were such cases observed or excluded? Additionally, did the authors measure or retrieve any evolution of the size distribution (e.g., dn/dlog Dp) as humidity changed? Please clarify.***

Response: Thank you for the valuable advice. The particle size range of 6–12 µm reported in the manuscript corresponds to the operational window within which our optical tweezers can stably trap droplets. When the particle size becomes too small, the optical gradient force is no longer sufficient to counteract scattering forces and Brownian motion, leading to unstable trapping or particle loss. Droplets smaller than this range were occasionally observed during dehumidification, but such cases were infrequent; therefore, they were excluded from analysis. Overall, the droplets that could be stably trapped and measured remained within the reported 6–12 µm range. This sentence has been added into this paragraph:

"Smaller particles become unstable in the trap as the optical gradient force is too weak, causing them to escape from the trap."

Additionally, the optical tweezers capture only a single particle at a time, our technique inherently cannot measure the evolution of an ensemble size distribution (e.g., dn/dlog Dp) during RH changes. Such distribution-level information is typically obtained using HTDMA-type instruments rather than single-particle optical tweezers. For droplets composed of the same solute, measurements performed on different individual particles indicate that their growth factors at a given RH agree within experimental uncertainty. However, the number of particles that can be measured is limited, and thus it is not feasible to derive a statistically meaningful size distribution.

Nevertheless, during each experiment we retrieve the particle radius by fitting its Raman spectrum (as shown in Fig. S4). Each individual fit introduces random uncertainty, and at a fixed RH, the retrieved radii exhibit a distribution that approximates a normal distribution centered on the true value. This behavior is consistent across all RH conditions and reflects the intrinsic fitting noise rather than physical variability in particle size.

***If I understand correctly, the study considers only soluble compounds, implying an internal mixing assumption. How would the method perform for externally mixed aerosols? In the presence of organic material or surfactants, additional effects may arise. While this may be beyond the current scope, such situations are common for pristine aerosols (e.g., sea spray). Could the authors comment or speculate, based on their experience and findings, on the applicability or limitations of the method in these cases?***

Response: Thank you for this constructive comment. Indeed, our method is currently

applicable only to internally mixed particles. This is because the optical tweezers can trap only liquid droplets, and the retrieval framework—based on refractive-index and density constraints—requires the particle to be homogeneous.

For externally mixed aerosols, insoluble inclusions may be present, leading to a heterogeneous refractive-index distribution. In such cases, both optical trapping stability and the spherical, homogeneous Mie scattering assumption may break down. For these types of particles, techniques such as HTDMA, or the development of Bessel-beam optical tweezers capable of trapping solid particles, would be more suitable for hygroscopicity measurements (Zhao et al., 2020).

For particles containing substantial organic material or surfactants, such as sea spray aerosols, liquid–liquid phase separation (LLPS) may occur at low RH. This would invalidate the standard Mie-fitting procedure, and additional models—such as core–shell Mie calculations—would be required to retrieve the radii and refractive indices of the individual phases before applying further thermodynamic constraints (Vennes and Preston, 2019). In contrast, if no LLPS occurs, changes in surface tension induced by organics are unlikely to affect the results, because Kelvin effects are negligible for micron-sized droplets.

Although a detailed treatment of these scenarios is beyond the scope of the present study, our experience suggests that the method could be extended in the future by incorporating more sophisticated optical models (e.g., core–shell Mie theory) as well as trapping techniques compatible with multiphase particles. The paragraphs below have been added into section 3.2:

"However, our method is currently applicable only to internally mixed particles. This is because the optical tweezers can trap only liquid droplets, and the retrieval framework requires the particle to be homogeneous. For externally mixed aerosols, insoluble inclusions may be present, leading to a heterogeneous refractive-index distribution. In such cases, both optical trapping stability and the spherical, homogeneous Mie scattering assumption may break down. For these types of particles, techniques such as HTDMA, or the development of Bessel-beam optical tweezers capable of trapping solid particles, would be more suitable for hygroscopicity measurements (Zhao et al., 2020). For particles containing substantial organic material or surfactants, liquid–liquid phase separation (LLPS) may occur at low RH. This would invalidate the standard Mie-fitting procedure, and additional models—such as core–shell Mie calculations—would be required to retrieve the radii and refractive indices of the individual phases before applying further thermodynamic constraints (Vennes and Preston, 2019). In contrast, if no LLPS occurs, changes in surface tension induced by organics are unlikely to affect the results, because Kelvin effects are negligible for micron-sized droplets.

Although a detailed treatment of these scenarios is beyond the scope of the present study, we suggest that the method could be extended in the future by incorporating more sophisticated optical models (e.g., core–shell Mie theory) as well as trapping techniques compatible with multiphase particles."

***The analysis assumes that uncertainties in measured particle radius and refractive index follow a normal distribution. What is the justification for this choice? Aerosol properties are typically lognormally distributed, and it is not clear that a normal distribution is the most appropriate representation of measurement uncertainty.***

*Please explain why the normal assumption was selected and whether alternative distributions were considered.*

Response: Thank you for this thoughtful question. The lognormal distribution commonly associated with aerosol properties refers to ensemble particle size distributions (e.g., dn/dlog Dp). In contrast, our analysis concerns the uncertainty arising from repeated retrievals of radius and refractive index for the same particle at a fixed RH. These uncertainties originate from multiple independent sources, including detector noise, spectral fitting residuals, and numerical approximations. Under such conditions, the resulting measurement errors are expected to approach a normal distribution. This assumption is further supported by our experimental observations. As shown in Fig. S4 and the figure below, repeated radius retrievals at stable RH yield distributions that are symmetric and well approximated by a normal distribution centered on the true value. Similar behavior is observed for refractive index retrievals. The sentence below has been added into section 2.3:

"Normal distribution is chosen due to the detection noise and stochastic fitting processes."

[Figure]

*The manuscript discusses only the real part of the refractive index, which governs scattering. What about absorption? Please comment on the imaginary part of the refractive index and whether it influences the retrievals or the applicability of the method.*

Response: Thank you for this insightful comment. In the present study, we focus on the real part of the refractive index because the investigated particles are primarily weakly absorbing system at the excitation wavelength. Therefore, the imaginary part of the refractive index is negligible and does not significantly influence the retrievals. In addition, strongly absorbing particles may absorb the trapping laser and undergo laser-induced heating, which can prevent stable trapping. From a methodological perspective,

the retrieval of particle radius and refractive index is based on fitting the positions of WGMs in the Raman spectra. These WGM resonance positions are highly sensitive to the particle radius and the real part of the refractive index, while they are largely insensitive to the imaginary part k (Preston and Reid, 2013). The influence of k is mainly reflected in the resonance linewidth rather than the resonance position; thus, its effect on the retrieved radius and real refractive index can be neglected. Consequently, absorption does not significantly affect the applicability of the present method.

*You present the functional dependence of the growth factor for each individual compound, but it is unclear how this approach extends to mixtures. How does the functional form behave for a binary mixture of the analyzed components, and how would the growth factor be derived in such cases? Please clarify the applicability of the method to mixed systems.*

Response: Thank you for the helpful suggestion. For binary mixtures (such as the ammonium sulfate–sodium chloride particles discussed in the manuscript), the functional relationship between growth factor and relative humidity remains valid. In the revised manuscript, we have added the fitted parameters for these mixtures to Table 1 and included the corresponding fitted GF–RH curves in Figure 4.

*The manuscript presents a schematic of the experimental setup. Would it be possible to include a photograph of the actual setup, at least in the Supplementary Information, to provide additional clarity and context?*

Response: Thank you for the helpful suggestion. We agree that including a photograph of the actual experimental setup can provide additional clarity and context. A photograph of the optical tweezer system has now been added to the Supplementary Information (Fig. S6).

[Figure]

Figure S6. Live-action image of the optical tweezers system. (a) Before the laser is turned on. (b) After the laser is turned on

*line 91: Please define (spell out) the abbreviation when it first appears, "WGMs".*

Response: Thank you for the suggestion. The abbreviation *WGMs* is introduced at its first occurrence in the manuscript (line 48).

*line 246: Figure 5: What is the BIC method?*

Response: Sorry for the confusion. The term "BIC method" should indeed refer to the RIC method (refractive-index-constrained retrieval method), which is described in detail in Section 2.3. We have corrected this in the revised manuscript.

**References:**
Preston, T. C. and Reid, J. P.: Accurate and efficient determination of the radius, refractive index, and dispersion of weakly absorbing spherical particle using whispering gallery modes, J. Opt. Soc. Am. B, 30, 2113-2122, https://doi.org/10.1364/JOSAB.30.002113, 2013
Vennes, B. and Preston, T. C.: Calculating and fitting morphology-dependent resonances of a spherical particle with a concentric spherical shell, J. Opt. Soc. Am. A, 36, 2089, https://doi.org/10.1364/JOSAA.36.002089, 2019.
Zhao, W., Cai, C., Zhao, G., Zhao, C.: Design of Bessel Beam Optical Tweezers for Single Particle Study, Acta Scientiarum Naturalium Universitatis Pekinensis, 56, 1031-1037, 10.13209/j.0479-8023.2020.090, 2020.

---

## Author Comment (AC2)

Response to Anonymous Referee #2

*This is a nice piece of work, I enjoy reading it. This work provides a new method to measure single particle hygroscopicity using optical tweezer. This new method is well validated against E-AIM theoretical values and other observations from previous studies. Better measurements of single particle hygroscopicity could not only help us improve aerosol-cloud-climate interactions in the climate models, can also help us measure PM pollution more precisely (Chen 2025). The manuscript is well written and structured. I only have a few minor comments to help further improve the article, detailed below.*

Response: Thanks for your valuable comments, which really helped improve the manuscript. Below, we will provide a detailed and point-by-point response to your comments. All the changes have been included in the latest manuscript.

*A few minor comments may help improve the discussion.*

*1) suggest dropping the abbreviation of AOT, could just simply call aerosol optical tweezer (refer to only tweezer later for simplicity), because AOT is commonly referred to as aerosol optical thickness in the community, therefore, to avoid confusion.*

Response: Thank you for the suggestion. In the revised manuscript, we have dropped the abbreviation "AOT" and consistently refer to the optical tweezers as "optical tweezers" to denote the aerosol optical tweezers.

*2) L32: I guess you want to say "diameter" growth factor (CF)?*

Response: Thank you for the comment. Yes, we are referring to the "diameter growth factor (GF)" in this context, and we have clarified the terminology in the revised manuscript to avoid confusion.

"During the measurement and characterization process, Hygroscopic growth is commonly quantified by the diameter growth factor (GF), mass growth factor ($GF_{mass}$), and the hygroscopicity parameter ($\kappa$), which link dry and humidified particle properties (Petters and Kreidenweis, 2007; Tang et al., 2019)."

*3) Please provide explanation of error bars in the figure captions.*

Response: Thanks for your suggestion. We have provided explanation of error bars in the figure captions and main text in the revised manuscript.

"Additionally, the error bars for the optical tweezers measurements represent the standard deviation, obtained either directly from the statistics of measurements or calculated through error propagation."

"Figure 2. Measurement of ammonium sulfate hygroscopicity. (a) Apparent molar volume of ammonium sulfate as a function of ionic strength. (b) Measured refractive index and corresponding values derived from the constrained solute mass. The error bars for the optical tweezers

measurements represent the standard deviation."

**4) I think adding some discussion of the limitation of this new method, some forward looking of how to improve it in future studies, and some perspective of potential applications in future, these would help further strengthen the article.**

Response: Thank you for this valuable suggestion. We have added a discussion in the revised manuscript (mainly in section 3.2 and 3.3) to highlight the limitations, potential improvements, and future applications of the method. Specifically:

"However, our method is currently applicable only to internally mixed particles. This is because the optical tweezers can trap only liquid droplets, and the retrieval framework requires the particle to be homogeneous. For externally mixed aerosols, insoluble inclusions may be present, leading to a heterogeneous refractive-index distribution. In such cases, both optical trapping stability and the spherical, homogeneous Mie scattering assumption may break down. For these types of particles, techniques such as HTDMA, or the development of Bessel-beam optical tweezers capable of trapping solid particles, would be more suitable for hygroscopicity measurements (Zhao et al., 2020). For particles containing substantial organic material or surfactants, liquid–liquid phase separation (LLPS) may occur at low RH. This would invalidate the standard Mie-fitting procedure, and additional models—such as core–shell Mie calculations—would be required to retrieve the radii and refractive indices of the individual phases before applying further thermodynamic constraints (Vennes and Preston, 2019). In contrast, if no LLPS occurs, changes in surface tension induced by organics are unlikely to affect the results, because Kelvin effects are negligible for micron-sized droplets.

Although a detailed treatment of these scenarios is beyond the scope of the present study, we suggest that the method could be extended in the future by incorporating more sophisticated optical models (e.g., core–shell Mie theory) as well as trapping techniques compatible with multiphase particles."

**References:**
Vennes, B. and Preston, T. C.: Calculating and fitting morphology-dependent resonances of a spherical particle with a concentric spherical shell, J. Opt. Soc. Am. A, 36, 2089, https://doi.org/10.1364/JOSAA.36.002089, 2019.
Zhao, W., Cai, C., Zhao, G., Zhao, C.: Design of Bessel Beam Optical Tweezers for Single Particle Study, Acta Scientiarum Naturalium Universitatis Pekinensis, 56, 1031-1037, 10.13209/j.0479-8023.2020.090, 2020.